# Convolutional Neural Networks Generalization Utilizing the Data Graph Structure

**Yotam Hechtlinger, Purvasha Chakravarti & Jining Qin**
Department of Statistics
Carnegie Mellon University
Pittsburgh, PA 15213, USA
{yhechtli,pchakrav,jiningq}@stat.cmu.edu

## Abstract

Convolutional Neural Networks have proved to be very efficient in image and audio processing. Their success is mostly attributed to the convolutions which utilize the geometric properties of a low - dimensional grid structure. This paper suggests a generalization of CNNs to graph-structured data with varying graph structure, that can be applied to standard regression or classification problems by learning the graph structure of the data. We propose a novel convolution framework approach on graphs which utilizes a random walk to select relevant nodes. The convolution shares weights on all features, providing the desired parameter efficiency. Furthermore, the additional computations in the training process are only executed once in the pre-processing step. We empirically demonstrate the performance of the proposed CNN on MNIST data set, and challenge the state-of-the-art on Merck molecular activity data set.

## 1 Introduction

Convolutional Neural Networks (CNNs) (LeCun et al., 1998) are variants of multi-layer perceptrons that have been inspired by biological cells in the visual cortex. The cells act as local filters over the input space and are well-suited to exploit the strong local spatial correlation present in natural images (Hubel & Wiesel, 1968). In recent years, following a breakthrough by Krizhevsky et al. (2012) at the 2012 ImageNet challenge, CNN has repeatedly demonstrated significant improvements in a large number of computer vision problems.

The major success of CNN for visual data is justly credited to the convolution. But its strength is dependent on three crucial underlying attributes found in visual data.

1. **Local connectivity assumption:** The signal in visual data tends to be highly correlated in local regions, and mostly uncorrelated in global regions.

2. **Shared weights assumption:** The same convolution is globally valid across the image, resulting in a significant parameter reduction.

3. **Grid structure of the image:** Enabling a straight forward re-scaling of the feature layers through the process of max pooling.

These assumptions make it challenging to duplicate the success of CNN on a different data structure. Nevertheless, CNNs have also proved effective for non-image data, usually relying on the grid structure of the inputs. Results on acoustic data (Hinton et al., 2012), videos (Le et al., 2011) and even Go board (Silver et al., 2016) indicate that it might be sensible to generalize CNN on other data structures that lack the under-lying grid structure.

The main contribution of this work is a generalization of CNNs to general graph-structured data, directed or undirected, offering a single method that incorporates the structural information present in the graph of the features into supervised learning algorithms. Due to the active research on learning the graph structure of features, this proves to be quite a general framework. As demonstrated by the examples, large number of standard continuous regression and classification problems fall within the

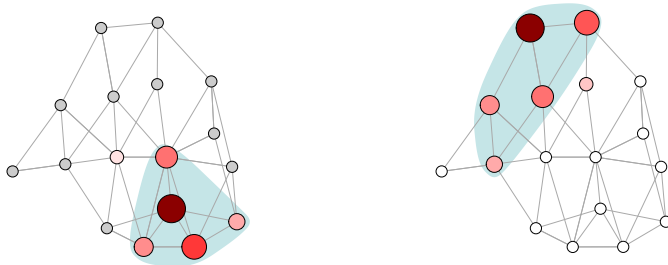

Figure 1: Visualization of the graph convolution size 5 . For a given node, the convolution is applied on the node and its 4 closest neighbors selected by the random walk. As the right figure demonstrates, the random walk can expand further into the graph to higher degree neighbors. The convolution weights are shared according to the neighbors' closeness to the nodes and applied globally on all nodes.

scope of this paper, by first estimating the graph structure of the data and then applying the proposed CNN on it.

The main hurdle for generalizing CNNs to graph-structured data is to find a corresponding generalized convolution operator. We first consider a random walk on the graph in order to select the top $k$ neighbors for every node during the pre-processing step, as Figure 1 shows. Then during the training process, the convolution is performed as an inner product of the weights and the selected top neighbors of the corresponding node in the preference order. Thus the weights are shared by each node and reflect the dependency between each node and its closest neighbors. When an image is considered as an undirected graph, this convolution operation is the same as the standard convolution. The proposed convolution is also applicable when the graph structure varies between observations.

In order to demonstrate the potential of the suggested method, we perform a set of experiments on the Merck molecular activity challenge and the MNIST data sets. The Merck molecular activity challenge data can be seen as a standard regression problem with significant correlation between the features. Essentially, for any regression or classification problem, the data can be visualized as a graph and its correlation matrix can be used to learn the corresponding graph structure. By treating the data as a graph, we show that a simple application of the graph convolutional neural network gives results that are comparable to state-of-the-art models.

## 2 LITERATURE REVIEW

Graph theory has been heavily studied in the last few decades, both from mathematical and statistical/computational perspectives, with a large body of algorithms developed for a variety of problems. Despite that, research on algorithms that incorporate CNNs with graph structured-data is still emerging. The idea of extending CNN to graph-structured data was recently explored by Bruna et al. (2013) and Henaff et al. (2015). They suggested two solutions. The first uses multi-scale clustering to define the network architecture, with the convolutions being defined per cluster without any weight sharing. The second defines the convolution through the eigen-values of the graph Laplacian, weighting out the distance induced by the graph's similarity matrix. The drawback of the methods is that there is no easy way to induce weight sharing among the different nodes of the graph. Also, these methods only handle inputs of a fixed size as the graph structure is fixed.

Standard CNN architectures use a fixed-dimensional input which makes it difficult to apply them on data with changing graph-structure. Recently, Kalchbrenner et al. (2014) developed a CNN for modeling sentences of varying lengths. Another interesting example of a convolution over a changing graph structure has recently been suggested by Duvenaud et al. (2015).

Several deep neural networks have been suggested in the past for predicting the properties of molecules (for example, Glen et al. (2006) and Lusci et al. (2013)). One of the proposed ideas is to extract features from the molecular structure into a fixed-dimensional feature vector and then use it

as an input in a machine learning method. Specifically, Duvenaud and Maclaurin Duvenaud et al. (2015), propose a neural network to extract features or molecular fingerprints from molecules that can be of arbitrary size and shape. Their neural network consists of layers which are local filters being applied to all the nodes and its neighbors. After using several such convolutional layers to create representations of the original data, they apply a global pooling step to features and feed that into a standard classifier. However, this method is limited in its ability to propagate information across the graph, limited by the depth of the network in its pooling stage.

The problem of selecting nodes for a convolution on a graph is a particular instance of the problem of selecting local receptive fields in a general neural network. The work of Coates & Ng (2011) suggest to select the local receptive fields in a general neural network according to closest neighbors induced by the similarity matrix.

In contrast to previous research, we suggest a novel efficient convolution that captures the local connectivity reflected in the graph structure. The convolution weights are shared among the different nodes and can even be applied to changing graph structures. We do so by considering the closest neighbors obtained in a random walk, using information contained in the similarity matrix.

## 3 GRAPH CONVOLUTIONAL NEURAL NETWORK

The key step which differentiates CNNs on images from regular neural networks, is the selection of neighbors on the grid in a $k \times k$ window combined with the shared weight assumption. We propose a convolution operator analogous to the convolution performed on images in standard CNNs. In order to select the local neighbors of a given node, we use the graph transition matrix and calculate the expected number of visits of a random walk starting from the given node. The convolution would then be applied on the nodes being visited the most. In this section we discuss the application of the convolution in a single layer on a single graph. It is immediate to extend the definition to more complex structures, and it will be explicitly explained in 3.4. We introduce some notation in order to proceed into further discussion.

**Notation:** Let $\mathcal{G} = (\mathcal{V}, \mathcal{E})$ be a graph over a set of $N$ features, $\mathcal{V} = (X_1, \ldots, X_N)$, and a set of edges $\mathcal{E}$. Let $P$ denote the transition matrix of a random walk on the graph, such that $P_{ij}$ is the probability to move from node $X_i$ to $X_j$. Let the similarity matrix and the correlation matrix of the graph be given by $S$ and $R$ respectively. Define $D$ as a diagonal matrix where $D_{ii} = \sum_j S_{ij}$.

### 3.1 TRANSITION MATRIX AND EXPECTED NUMBER OF VISITS

This work assumes the existence of the graph transition matrix $P$. This is not a restriction. If graph structure of the data is already known, i.e. if the similarity matrix $S$ is already known, then the transition matrix can be obtained, as explained in Lovász et al. (1996), by

$$P = D^{-1}S. \tag{1}$$

If the graph structure is unknown, it can be learned using several unsupervised or supervised graph learning algorithms. Learning the data graph structure is an active research topic and is not in the scope of this paper. The interested reader can start with Belkin & Niyogi (2001), and Henaff et al. (2015) discussing similarity matrix estimation. We use the absolute value of the correlation matrix as the similarity matrix, following Roux et al. (2008) who showed that correlation between the features is usually enough to capture the geometrical structure of images. That is, we assume

$$S_{i,j} = |R_{i,j}| \ \forall \, i, j. \tag{2}$$

Once we derive the transition matrix $P$, we define $Q^{(k)} := \sum_{i=0}^{k} P^k$, where $[P^k]_{ij}$ is the probability of transitioning from $X_i$ to $X_j$ in $k$ steps. That is,

$$Q^{(0)} = I, \ \ Q^{(1)} = I + P, \cdots, Q^{(k)} = \sum_{i=0}^{k} P^k. \tag{3}$$

Note that $Q_{i,j}^{(k)}$ is also the expected number of visits to node $X_j$ starting from $X_i$ in $k$ steps. The $i^{th}$ row, $Q_{i,\cdot}^{(k)}$ provides a measure of similarity between node $X_i$ and its neighbors by considering

a random walk on the graph. As $k$ increases we incorporate neighbors further away from the node, while the act of summation still gives proper weights to the node itself and its closest neighbors. Figure 2 provides a visualization of the matrix $Q$ over the 2-D grid.

To the best of our knowledge, this is the first use of the expected number of visits on a graph to select neural nets architecture. Coates & Ng (2011) and others suggest using the similarity matrix. This definition extends the notion of the similarity matrix, since $Q^{(1)}$ agrees with the variable order induced by the similarity matrix. Furthermore, higher powers of $k$ emphasize more on the graph structure of the data, giving major hubs more weight. This might be valuable, for example, in social network data.

## 3.2 CONVOLUTIONS ON GRAPHS

As discussed earlier, each row of $Q^{(k)}$ can be used to obtain the closest neighbors of a node. Hence it seems natural to define the convolution over the graph node $X_i$ using the $i$'th row of $Q^{(k)}$. In order to do so, we denote $\pi_i^{(k)}$ as the permutation order of the $i^{th}$ row of $Q^{(k)}$ in descending order. That is, for every $i = 1, 2, ..., N$ and every $k$,

$$\pi_i^{(k)} : \{1, 2, ..., N\} \longrightarrow \{1, 2, ..., N\},$$

such that $Q_{i,\pi_i^{(k)}(1)} > Q_{i,\pi_i^{(k)}(2)} > ... > Q_{i,\pi_i^{(k)}(N)}$.

The notion of ordered distance between the nodes is a global feature of all graphs and nodes. Therefore, we can take advantage of it to satisfy the desired shared weights assumption. We define $Conv_1$, as the size $p$ convolution over the graph $G$ with nodes $\mathbf{x} \in R^N$ and weights $\mathbf{w} \in R^p$, for the $p$ nearest neighbors of each node, as the inner product:

$$Conv_1(\mathbf{x}) = \begin{bmatrix} x_{\pi_1^{(k)}(1)} & \cdots & x_{\pi_1^{(k)}(p)} \\ x_{\pi_2^{(k)}(1)} & \cdots & x_{\pi_2^{(k)}(p)} \\ \vdots & \ddots & \vdots \\ x_{\pi_N^{(k)}(1)} & \cdots & x_{\pi_N^{(k)}(p)} \end{bmatrix} \cdot \begin{bmatrix} w_1 \\ w_2 \\ \vdots \\ w_p \end{bmatrix}, \quad \text{where } \mathbf{x} = \begin{bmatrix} x_1 \\ x_2 \\ \vdots \\ x_N \end{bmatrix} \qquad (4)$$

The order of the weights follows from the distance induced by the transition matrix. That is, $w_1$ will be convoluted with the variable which has the largest value in each row according to the matrix $Q^{(k)}$. For example, when $Q^{(1)} = I + P$, $w_1$ will always correspond to the node itself, and $w_2$ will correspond to the node's closest neighbor. For higher values of $k$, the order will be defined by the unique graph structure. An interesting attribute of this convolution, as compared to other convolutions on graphs is that, it preserves locality while still being applicable over different graphs with different structures.

It should be noted that $Conv_1$ is susceptible to the effects of negative correlation between the features, and does not take into account the actual distance between the nodes (it only uses that for the selection of the closest neighbors of a node). Since the weights are being learned globally, in order to account for that, we have also defined $Conv_2$ as:

$$Conv_2(\mathbf{x}) = \begin{bmatrix} y_{1,\pi_1^{(k)}(1)} & \cdots & y_{1,\pi_1^{(k)}(p)} \\ y_{2,\pi_2^{(k)}(1)} & \cdots & y_{2,\pi_2^{(k)}(p)} \\ \vdots & \ddots & \vdots \\ y_{N,\pi_N^{(k)}(1)} & \cdots & y_{N,\pi_N^{(k)}(p)} \end{bmatrix} \cdot \begin{bmatrix} w_1 \\ w_2 \\ \vdots \\ w_p \end{bmatrix}, \qquad (5)$$

$$\text{where } \mathbf{x} = \begin{bmatrix} x_1 \\ x_2 \\ \vdots \\ x_N \end{bmatrix} \text{ and } y_{ij} = \text{sign}(R_{ij}) \, Q_{ij} \, x_j, \, \forall \, i = 1, ..., N, \, j = 1, ..., N.$$

In practice $Conv_2$ performs slightly better than $Conv_1$, although the major differences between them are mostly smoothed out during the training process.

An important feature of the suggested convolution is the operation complexity. For a graph with $N$ nodes, a single $p$ level convolution only requires $O(N \cdot p)$ operations, where $p$ is a very small

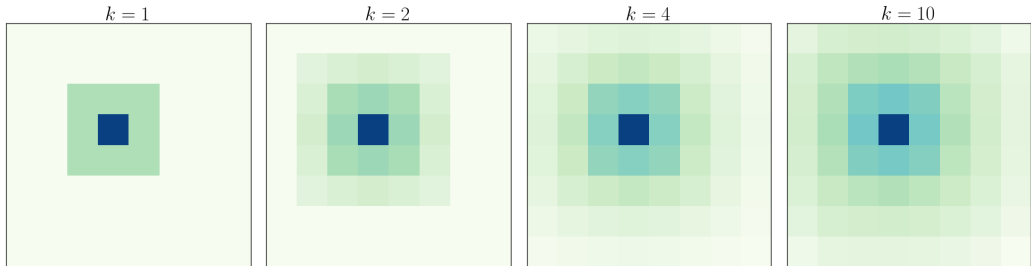

Figure 2: Visualization of a row of $Q^{(k)}$ on the graph generated over the 2-D grid at a node near the center, when connecting each node to its 8 adjacent neighbors. For $k = 1$, most of the weight is on the node, with smaller weights on the first order neighbors. This corresponds to a standard $3 \times 3$ convolution. As $k$ increases the number of active neighbors also increases, providing greater weight to neighbors farther away, while still keeping the local information.

natural number (the number of neighbors considered). The major computational effort goes in the computation of $Q$ which is being done once per graph structure in the pre-processing step.

### 3.3 SELECTION OF THE POWER OF Q (K)

The selection of the value of $k$ is data dependent, as with every hyper-parameter. But there are two main components affecting its value. Firstly, it is necessary for $k$ to be large enough to detect the top $p$ neighbors of every node. If the transition matrix $P$ is sparse, it might require higher values of $k$. Secondly, from properties of stochastic processes, we know that if we denote $\pi$ as the Markov chain stationary distribution, then

$$\lim_{k \to \infty} \frac{Q_{ij}^{(k)}}{k} = \pi_j \ \ \forall \, i, j. \tag{6}$$

This implies that for large values of $k$, local information will be smoothed out and the convolution will repeatedly be applied on the features with maximum connections. For this reason, we suggest the value of $k$ to be kept relatively low (but large enough to capture sufficient amount of features, when needed).

### 3.4 IMPLEMENTATION

Similar to standard convolution implementation (Chellapilla et al., 2006), it is possible to represent the graph convolution as a tensor dot product, transferring most of the computational burden to the GPU while using highly optimized matrix product libraries.

For every graph convolution layer, we have as an input a $3D$ tensor of observations, their features and depth. We first extend the input with an additional dimension that includes the top $p$ neighbors of each feature selected by $Q^{(k)}$, transforming the input dimension from $3D$ to $4D$ tensor as

$$(\text{Observations, Features, Depth}) \rightarrow (\text{Observations, Features, Neighbors, Depth}) \,.$$

Now for every graph convolution layer, the weights are a $3D$ tensor with the dimension of (Neighbors, Depth, Filters). Therefore application of a graph convolution which is a tensor dot product between the input and the weights, along the (Neighbors, Depth) axes, results with an output dimension:

$$\Big( (\text{Observations, Features}) \,, (\text{Neighbors, Depth}) \Big) \bullet \Big( (\text{Neighbors, Depth}) \,, (\text{Filters}) \Big)$$
$$= (\text{Observations, Features, Filters}) \,.$$

Implementation of the algorithm has been done using Keras and Theano libraries in Python, inheriting all the tools provided by the libraries to train neural networks, such as dropout regularization, advanced optimizers and efficient initialization methods. Source code will be publicly available prior to the ICLR conference on the authors' website.

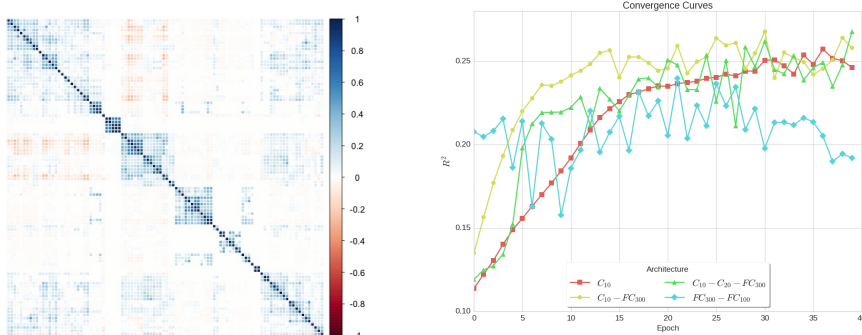

Figure 3: **Left:** Visualization of the correlation matrix between the first 100 molecular descriptors (features) in the DPP4 Merck molecular activity challenge training set. The proposed method utilizes the correlation structure between the features. **Right:** Convergence of $R^2$ for the different methods on the test set. The graph convolution makes the convergence steadier by reducing the number of parameters.

## 4 EXPERIMENTS

In order to test the feasibility of the proposed CNN on graphs, we have conducted experiments on well known data sets functioning as benchmarks - Merck molecular activity challenge and MNIST. Both the data sets are popular and well-studied challenges in computational biology and computer vision respectively.

In all the implementations we kept the architecture shallow and simple, instead of deep and complex. This was done to enable better comparisons between the models, and reduce the chance of over-fitting the test set by the model selection process. The hyper-parameters were chosen arbitrarily when possible rather than being tuned and optimized. Nevertheless, we still report state-of-the-art, or competitive results on the experimented data sets.

In this section, we denote a graph convolution layer with $k$ feature maps by $C_k$ and a fully connected layer with $k$ hidden units by $FC_k$.

### 4.1 MERCK MOLECULAR ACTIVITY CHALLENGE

The Merck molecular activity is a Kaggle [1] challenge which is based on 15 molecular activity data sets. The target is to predict activity levels for different molecules based on the structure between the different atoms in the molecule. This helps to identify molecules in medicines which hit the intended target and do not cause side effects.

Following Henaff et al. (2015), we apply our algorithm on the DPP4 dataset. DPP4 contains 6148 training and 2045 test molecules. Some of the features of the molecules are very sparse and are only active in a handful number of molecules. For these features, the correlation estimation is not very accurate. Therefore we use features that are active in at least 20 molecules (observations). This results in 2153 features. As can be seen in Figure 3, there is significant correlation structure between different features. This implies strong connectivity among the features which is important for the application of the proposed method.

The training was performed using Adam optimization procedure (Kingma & Ba, 2014) where the gradients are derived from back-propagation algorithm. We used learning rate, $\alpha = 0.001$, fixed the number of epochs to 40 and implemented dropout regularization on every layer during the optimization procedure. The correlation matrix absolute values were used to learn the graph structure. We found that a small number of nearest neighbors ($p$) between 5 to 10 works the best, and used $p = 5$ in all models.

Since this is a regression problem, we used the root mean-squared error loss (RMSE). Following the standard set by the Kaggle challenge, results are reported in terms of the squared correlation ($R^2$),

---

[1]Challenge website is `https://www.kaggle.com/c/MerckActivity`

| Method | Architecture | $R^2$ |
|---|---|---|
| OLS Regression | | 0.135 |
| Random Forest | | 0.232 |
| Merck winner DNN | | 0.224 |
| Spectral Networks | $C_{64}$- $P_8$ - $C_{64}$ - $P_8$ - $FC_{1000}$ | 0.204 |
| Spectral Networks (supervised graph) | $C_{16}$- $P_4$ - $C_{16}$ - $P_4$ - $FC_{1000}$ | 0.277 |
| Fully connected NN | $FC_{300}$-$FC_{100}$ | 0.192 |
| Graph CNN | $C_{10}$ | 0.246 |
| Graph CNN | $C_{10}$- $FC_{100}$ | 0.258 |
| Graph CNN | $C_{10}$- $C_{20}$- $FC_{300}$ | 0.268 |

Table 1: The squared correlation between the actual activity levels and predicted activity levels, $R^2$ for different methods on DPP4 data set from Merck molecular activity challenge.

that is,

$$R^2 = \text{Corr}(Y, \hat{Y})^2,$$

where $Y$ is the actual activity level and $\hat{Y}$ is the predicted one.

The convergence plot given in Figure 3 demonstrates convergence of the selected architectures. The contribution of the suggested convolution is explained in view of the alternatives:

- **Fully connected Neural Network:** Models first applying convolution, followed by fully connected hidden layer converge better than more complex fully connected models. Furthermore, convergence is more stable in comparison to the fully connected models, due to the parameter reduction.

- **Linear Regression:** Optimizing over the set of convolutions is often considered as automation of the feature extraction process. From that perspective, a simple application of one layer of convolution, followed by linear regression, significantly outperforms the results of a standalone linear regression.

Table 1 provides more thorough $R^2$ results for the different architectures explored, and compares it to two of the winners of the Kaggle challenge, namely the Deep Neural Network and the Random forest in Ma et al. (2015). We perform better than both the winners of the Kaggle contest. Since the challenge is already over, and we had full access to the test set, the results should mostly be considered as a proof of concept.

The models in Henaff et al. (2015) and Bruna et al. (2013) use a spectral approach, and currently hold the state-of-the-art. In comparison to them, we perform better than the Spectral Networks CNN on unsupervised graph structure, which is equivalent to what was done by using the correlation matrix as similarity matrix. The one using Spectral Networks on supervised graph structure holds the state-of-the-art by learning the graph structure. This is a direction we have not yet explored, as graph learning is beyond the scope of this paper, although it will be straightforward to apply the proposed graph CNN in a similar way to any learned graph.

## 4.2 MNIST DATA

The MNIST data often functions as a benchmark data set to test new machine learning methods. We have experimented with two different graph structures for the images. First, we considered the images as observations from an undirected graph on the 2-D grid, where each pixel is connected to its 8 adjunct neighbors pixels. We used the convolutions over the grid structure as presented in Figure 2, and $Q^{(3)}$ with $p = 25$ as the number of nearest neighbors. Due to the symmetry of the graph in most regions of the image, many pixels has equal distance from the pixel being convoluted. If ties were broken in a consistent manner, this example would be reduced to the regular convolution on a $5 \times 5$ window for exactly the entire space but pixels 3 steps away from the boundary. In order to make the example more compelling, we have broken ties arbitrarily, making the training process harder compared to regular CNN. Imitating LeNet, with $C_{20}, Pooling_{(2\times2)}, C_{50}, Pooling_{(2\times2)}, FC_{100}$

| Method | Error Rate (%) | # of Parameteres |
|---|---|---|
| Logistic Regression | 7.49 | $7,180$ |
| $C_{20}$ | 2.24 | $143,410$ |
| $C_{20} - C_{20}$ | 1.71 | $145,970$ |
| $C_{20} - FC_{512}$ | 1.39 | $7,347,862$ |
| $FC_{512} - FC_{512}$ | 1.42 | $635,402$ |

Table 2: Error rates of different methods on MNIST digit recognition task.

followed by a linear classifier, resulted with $1.1\%$ error rate. This is a worse than a regular CNN which achieves with similar architecture around $0.75\%$ error rate, and better than a fully connected neural network which achieves around $1.4\%$, as expected from the complexity differences of the models.

Second, we used the correlation matrix to estimate the graph structure directly from the pixels. Since some of the MNIST pixels are constant (e.g the corners are always black), we restricted the data only to the active 717 pixels not constant. We used $Q^{(1)}$ with $p = 6$ as the number of neighbors. This was done in order to ensure that the spatial structure of the image no longer effect the results. With only 6 neighbors, and a partial subset of the pixels, the relative location of the top correlated pixels necessary varies per pixel. As a result, regular CNN are no longer applicable on the data, and we have compared the performance to fully connected Neural Networks.

Table 2 present the experiment results. During training a dropout rate of $0.2$ has been applied on all layers to prevent overfitting. In all the experiments the final layer is the standard softmax logistic regression classifier. The Graph CNN perform on par with fully connected neural networks, with lower number of parameters. Also a single layer of graph convolution, followed by logistic regression greatly improve the performance of logistic regression, demonstrating the potential of the graph convolution for feature extraction purposes. As with regular convolutions, $C_{20} - FC_{512}$ had over $7M$ parameters, due to the fact that the convolution uses small amount of parameters to generate different maps of the input. This implies that the graph convolution might be even more effective with the development of an efficient pooling methods on graphs, a problem that will be covered in future research.

## 5 CONCLUSIONS

We suggest a method to address the problem of supervised learning over graph-structured data, by extending convolutional neural networks to graph input. Our main contribution is a new way to define a convolution over graph that can handle different graph structures as its input. The convolution can be applied on standard regression or classification problems by learning the graph structure in the data, using the correlation matrix, or other methods. Compared to a fully connected layer, the suggested convolution has significantly lower number of parameters, while providing stable convergence and comparable performance. We validated and demonstrated the predictive performance of our proposed method on benchmark machine learning data sets such as: the Merck Molecular Activity data set and MNIST data.

Convolutional Neural Networks have already revolutionized the field of computer vision, speech recognition and language processing. We think an important step forward is to extend it to all other problems which have an inherent graph structure within them.

### ACKNOWLEDGMENTS

We would like to thank Alessandro Rinaldo, Ruslan Salakhutdinov and Matthew Gormley for suggestions, insights and remarks that has greatly improved the quality of this paper.

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
