# Peer review of "Convolutional Neural Networks Generalization Utilizing the Data Graph Structure"

_ICLR 2017 — rejected_

[Public Comment · Tara N Sainath · 07 Nov 2016]
**ICLR Paper Format**

Dear Authors,

Please resubmit your paper in the ICLR 2017 format stating in the header "submitted" instead of "published" for your submission to be considered. Thank you!

[Official Review · AnonReviewer2 · rating 3 · confidence 3 · 16 Dec 2016]
**Important problem, but lacks clarity and I'm not sure what the contribution is.**

This work proposes a convolutional architecture for any graph-like input data (where the structure is example-dependent), or more generally, any data where the input dimensions that are related by a similarity matrix. If instead each input example is associated with a transition matrix, then a random walk algorithm is used generate a similarity matrix.

Developing convolutional or recurrent architectures for graph-like data is an important problem because we would like to develop neural networks that can handle inputs such as molecule structures or social networks. However, I don't think this work contributes anything significant to the work that has already been done in this area. 

The two main proposals I see in this paper are:
1) For data associated with a transition matrix, this paper proposes that the transition matrix be converted to a similarity matrix. This seems obvious.
2) For data associated with a similarity matrix, the k nearest neighbors of each node are computed and supply the context information for that node. This also seems obvious.

Perhaps I have misunderstood the contribution, but the presentation also lacks clarity, and I cannot recommend this paper for publication. 

Specific Comments:
1) On page 4: "An interesting attribute of this convolution, as compared to other convolutions on graphs is that, it preserves locality while still being applicable over different graphs with different structures."  This is false; the other proposed architectures can be applied to inputs with different structures (e.g. Duvenaud et. al., Lusci et. al. for NN architectures on molecules specifically).

[Official Review · AnonReviewer1 · rating 6 · confidence 3 · 17 Dec 2016 (modified: 23 Jan 2017)]

Update: I thank the authors for their comments! After reading them, I decided to increase the rating.

This paper proposes a variant of the convolution operation suitable for a broad class of graph structures. For each node in the graph, a set of neighbours is devised by means of random walk (the neighbours are ordered by the expected number of visits). As a result, the graph is transformed into a feature matrix resembling MATLAB’s/Caffe’s im2col output. The convolution itself becomes a matrix multiplication. 

Although the proposed convolution variant seems reasonable, I’m not convinced by the empirical evaluation. The MNIST experiment looks especially suspicious. I don’t think that this dataset is appropriate for the demonstration purposes in this case. In order to make their method applicable to the data, the authors remove important structural information (relative locations of pixels) thus artificially increasing the difficulty of the task. At the same time, they are comparing their approach with regular CNNs and conclude that the former performs poorly (and does not even reach an acceptable accuracy for the particular dataset).

I guess, to justify the presence of MNIST (or similar datasets) in the experimental section, the authors should modify their method to incorporate additional graph structure (e.g. relative locations of nodes) in cases when the relation between nodes cannot be fully described by a similarity matrix.

I believe, in its current form, the paper is not yet ready for publication but may be later resubmitted to a workshop or another conference after the concern above is addressed.

[Reviewer Comment · AnonReviewer3 · rating 3 · 19 Dec 2016]
**Modifies the way neighbors are computed for Graph-convolutional networks, but doesn't show that this modification is an improvement..**

Previous literature uses data-derived adjacency matrix A to obtain neighbors to use as foundation of graph convolution. They propose extending the set of neighbors by additionally including nodes reachable by i<=k steps in this graph. This introduces an extra tunable parameter k, so it needs some justification over the previous k=1 solution. In one experiment provided (Merk), using k=1 worked better. They don't specify which k that used, just that it was big enough for their to be p=5 nodes obtained as neighbors. In the second experiment (MNIST), they used k=1 for their experiments, which is what previous work (Coats & Ng 2011) proposed as well. A compelling experiment would compare to k=1 and show that using k>1 gives improvement strong enough to justify an extra hyper-parameter.

[Final Decision · Program Chairs · 06 Feb 2017]
**ICLR committee final decision**

This work studies the problem of generalizing a convolutional neural network to data lacking grid-structure. 
 
 The authors consider the Random Walk Normalized Laplacian and its finite powers to define a convolutional layer in a general graph. Experiments in Merck molecular discovery and mnist are reported. 
 
 The reviewers all agreed that this paper, while presenting an interesting and important problem, lacks novelty relative to existing approaches. In particular, the AC would like to point out that important references seem to be missing from the current version. 
 
 The proposed approach is closely related to 'Convolutional neural networks on graphs with fast localized spectral filtering', Defferrand et al. NIPS'16 , which considers Chevyshev polynomials of the Laplacian and learns the polynomial coefficients in an efficient manner. Since the Laplacian and the Random Walk Normalized Laplacian are similar operators (have same eigenvectors), the resulting model is essentially equivalent. Another related model that precedes all the cited works and is deeply related to the current submission is the Graph Neural Network from Scarselli et al.; see 'Geometric Deep Learning: going beyond Euclidean Data', Bronstein et al,